# Plant Diversity and Species Composition in Relation to Soil Enzymatic Activity in the Novel Ecosystems of Urban–Industrial Landscapes

Wojciech Bierza [1], Joanna Czarnecka [2,*], Agnieszka Błońska [1], Agnieszka Kompała-Bąba [1], Agnieszka Hutniczak [1], Bartosz Jendrzejek [1], Jawdat Bakr [1,3], Andrzej M. Jagodziński [4,5], Dariusz Prostański [6] and Gabriela Woźniak [1,*]

1  Institute of Biology, Biotechnology and Environmental Protection, Faculty of Natural Sciences, University of Silesia in Katowice, 28 Jagiellońska Str., 40-032 Katowice, Poland
2  Department of Botany, Mycology and Ecology, Institute of Biological Sciences, Maria Curie-Skłodowska University, 19 Akademicka Str., 20-033 Lublin, Poland
3  Technical Institute of Bakrajo, Sulaimani Polytechnic University SPU, Qrga Wrme Str.-327/76, Sulaymaniyah 46001, Iraq
4  Department of Ecology, Institute of Dendrology, Polish Academy of Sciences, 5 Parkowa Str., 62-035 Kórnik, Poland
5  Department of Game Management and Forest Protection, Faculty of Forestry and Wood Technology, Poznań University of Life Sciences, 71D Wojska Polskiego Str., 60-625 Poznań, Poland
6  KOMAG Institute of Mining Technology, 37 Pszczynska Str., 44-101 Gliwice, Poland
*  Correspondence: joanna.czarnecka@mail.umcs.pl (J.C.); gabriela.wozniak@us.edu.pl (G.W.)

**Abstract:** The primary producers and processes of matter and energy flow, reflected by the soil enzyme activity, are the basics of all ecosystem functioning processes. This paper reviews the relationships between the plant diversity, the physicochemical substrate parameters, and the soil enzymatic activity in novel ecosystems of the urban–industrial landscape, where the factors driving soil enzyme activity are not fully understood and still need to be studied. The relationship between the biotic and abiotic factors in the development of novel ecosystems on de novo established habitats, e.g., sites of post-mineral excavation, are shaped in ways unknown from the natural and the semi-natural habitats. The main criteria of de novo established ecosystems are the vegetation patches of the non-analogous species composition created as a result of human impact. The non-analogous species assemblages are associated with different microorganism communities because the biomass and the biochemistry of soil organic matter influence the enzyme activity of soil substrates. Moreover, the soil enzyme activity is an indicator that can dynamically reflect the changes in the microbial community structure dependent on the best-adapted plant species, thanks to the particular traits and individual adaptive adjustments of all the plant species present. This way, soil enzyme activity reflects the sum and the interactions of the elements of the ecosystem structure, irrespective of the vegetation history and the habitat origin.

**Keywords:** soil dehydrogenase; soil phosphatase; soil substratum; physicochemical parameters; ruderal sites; vegetation; coal mine heaps

## 1. Introduction

In natural and semi-natural ecosystems, the soil organic matter (SOM) is derived primarily from autotrophic organisms, mostly plant species [1–5]. Some amount of the SOM also comes from animal tissue and manures [6,7]. The occurrence of any heterotrophic organisms in the soil, e.g., microorganisms and soil fauna, depends on the type of organic matter. The character of the soil organic matter reflects the unique compounds synthesized by the autotrophic organisms, primarily by plants [8–10]. The quantity and quality of the produced biomass depend on the variety of vegetation characteristics (e.g.,

the plant species composition, the plant diversity, and the chemical composition of the produced biomass). The relationship between the habitat's abiotic conditions and the biochemistry of the above- and below-ground plant biomass are the crucial elements that further shape the development of the habitat's characteristics, and they are the fundamental factors influencing the processes of the establishment and functioning of the natural and seminatural ecosystems [11–13]. The amount and characteristics of the plant above- and below-ground biomass depend on various factors, including vegetation dominated by annual or perennial species, soil substratum parameters, such as moisture (especially in conditions of drought) [14], nutrient content, texture, soil development stage, and other soil biogeochemical and physical parameters [15,16]. Climate and weather conditions are further significant factors to consider in this context. The organic matter decomposition, nutrient cycling, and energy flow in the natural and the semi-natural ecosystems all depend on the biological components of plants, bacteria, fungi, protozoa, invertebrates (e.g., insects, earthworms, mites, nematodes), and vertebrates. They all play a vital role in maintaining the soil structure and function [10], and the functional traits of these organisms are also significant [17,18].

The network characterized above remains in a dynamic balance. However, it is changed and transformed significantly when human activity is introduced. Our concept about the differences in the impact on environmental functioning among varied levels of human activities is presented in Figure 1. The fast economic development of numerous countries takes place on the increasing area of land covered by cities and industry. The spatial extent of man-made landscapes has been steadily increasing and occupies three-quarters of the Earth's land surface [19]. Human activity causes strong disturbances and stress, and the parallel dynamic adaptation processes of living organisms lead to the formation of altered ecosystems with non-analogous species composition. Habitat transformation is sometimes so substantial (e.g., sites of post-mineral excavation) that the emerging system resulting from the natural successional processes meets the set of criteria that defines novel ecosystems [20–22]. During our study, it was recorded that the vegetation composition underwent significant changes, reflecting the dynamic changes in the conditions of the abiotic habitats. Relationships between the abiotic and biotic parameters are scarcely studied in post-industrial habitats, and the main feature of the biotic background is plant non-analogous species composition [23,24]. Examples of novel ecosystems are the urban–industrial landscapes, the post-mineral excavation habitats, antique buildings, stonewalls, etc. [25–32]. Our study, conducted on plant species diversity and related microbial activity, as reflected in decomposition and other ecosystem processes on sites that underwent the excavation of mineral resources, inspired us to prepare this paper.

The objective of this paper is to review the relationship between the plant species composition, the diversity of novel ecosystems, and the soil/substratum abiotic and biotic characteristics, with special attention paid to enzymatic activity occurring in novel ecosystems in the urban–industrial landscapes. From this point of view, the paper presents a fundamentally new approach to functional ecosystems as the basis of ecosystem services. In particular, we would like to present (i) the relations between the substrate physicochemical properties filtering the plant diversity and the species composition and the soil enzyme activity in the novel ecosystems, and (ii) the relations between vegetation and soil enzyme activity in the novel ecosystems.

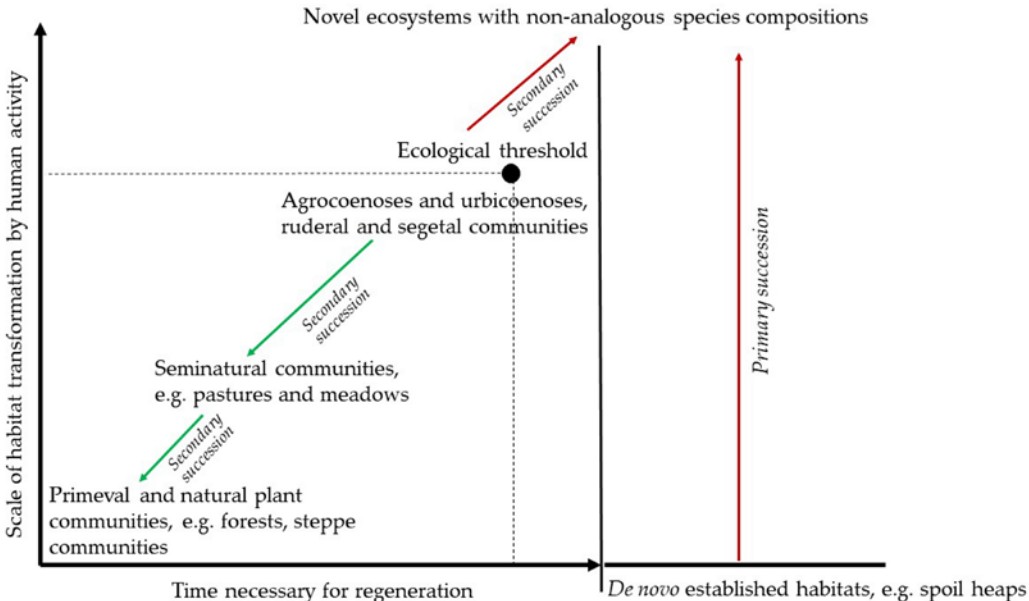

**Figure 1.** The transformations and resulting habitat types composing the mosaic of plant assemblages with possible successional directions when human influence ends.

## 2. Procedure for Selecting the Bibliographic Portfolio and Data Collection

After setting the research objective mentioned below, a systematic review of the published literature was carried out, searching on the SCOPUS database to identify relevant publications that studied three topics: (i) aspects of soil microbial activity in novel ecosystems, (ii) soil microbial activity in the urban areas, and (iii) relations between the soil microbial activity and the plant diversity. The search was limited to scientific articles and reviews. The search of the database took place during September 2022, using the following search queries:

(i)    TITLE-ABSTRACT-KEYWORDS ({soil microbial activity} OR {soil enzymes} OR {soil enzyme activity} AND {novel ecosystems} AND LIMIT-TO (LANGUAGE, "English") AND (LIMIT-TO (DOCTYPE, "article") OR LIMIT-TO (DOCTYPE, "review")).
(ii)   TITLE-ABSTRACT-KEYWORDS ({soil microbial activity} OR {soil enzymes} OR {soil enzyme activity} AND {urban area} OR {post industrial area} OR {spoil heaps} AND LIMIT-TO (LANGUAGE, "English") AND (LIMIT-TO (DOCTYPE, "article") OR LIMIT-TO (DOCTYPE, "review")).
(iii)  TITLE-ABSTRACT-KEYWORDS ({soil microbial activity} OR {soil enzymes} OR {soil enzyme activity} AND {plant diversity}) AND LIMIT-TO (LANGUAGE, "English") AND (LIMIT-TO (DOCTYPE, "article") OR LIMIT-TO (DOCTYPE, "review")).

English was selected as the language for the search. Topics (ii) and (iii) included papers from the publication years of 2000–2021, while topic (i) was limited to the publication year period of 2016–2021 due to a lack of previous papers.

Data for the presentation effects of the different plant species on the soil enzyme activity in the novel ecosystems were collected from tables demonstrated in articles where possible, or from figures using Web Plot Digitizer 4.60 [33].

## 3. Soil Enzyme Activity—Basic Information

Soil enzymes catalyze many reactions that are necessary in the processes related to the functioning of the soil microorganisms [34,35]. They are essential for stabilizing the soil structure, the decomposition of organic residues, the formation of organic matter, and the circulation of nutrients [10]. This explains why they are crucial to understanding the soil biochemical processes; soil biochemistry plays a fundamental role in understanding the global carbon cycle [35,36]. Plant roots, fungi, and bacteria secrete enzymes into the

soil, and they can also be identified as independent extracellular proteins. Apart from the living organisms releasing different proteins, the activity of enzymes is related to the soil physicochemical properties, the characteristics of the soil organic matter, and the composition as well as biomass of the soil microorganisms [37]. The measurements of the activity of the soil enzymes provide information on the functions of the soil microorganism community and their specific metabolic activities. It also reflects the natural and the human-induced disturbances in the analyzed ecosystem [38,39]. Measurements of the activity of the soil enzymes are applied to evaluate the intensity of the soil processes (such as impacts of drought and salinity), particularly the influence of the vegetation composition on the soil microbiological activity.

Vegetation composition can directly or indirectly change the soil properties because of the chemical and biochemical characteristics of the biomass. The close connection between the plants and the microorganisms is the fundamental factor of soil fertility [40]. Study results indicate that the composition of the microorganisms in soil may vary depending on the plant species and their abundance and the length of vegetation persistence [41–50]. Plant species produce different quantities of litter with different lignin, calcium, and nitrogen contents [51]. Strong relationships between the vegetation characteristics resulting from species composition and soil enzyme activity have been reported from semi-natural alpine meadows. The features of vegetation determine the structure of plant biomass remnants, which are composed of: (i) resistant structural compounds such as lignin, polyphenols, lipids, and cutin; (ii) moderately resistant structural compounds such as cellulose and hemicellulose; and (iii) highly labile intracellular compounds such as protein, starch, fructan, chlorophyll, and other pigments [52]. A study conducted on sandy soils revealed that the characteristics of vegetation determine the carbon sources fundamental to the metabolic activity of the microorganisms, which shape the functional diversity of the microbial assemblages [53]. Even different organs (i.e., leaves, stems, and coarse and fine roots) isolated from one individual plant decompose at different rates due to their different susceptibility to microbial utilization. The intensity of the decomposition of plant biomass differs in response to the proportions of various structural and intracellular compounds [54]. The differences in microorganism communities also depends on the number of fine roots and their metabolic activity, specifically for a particular plant species [55]. The second crucial factor is the chemical composition of the biomass and consequently the litter produced under the influence of the dominance of different plant species, as it influences the activity of the enzymes involved in the mineralization of the nutrients [43]. The quality and quantity of the biomass and litter are reflected in significant differences in the soil pH and nutrient cycling because of the activity of the microorganisms and the soil enzymes. The lignin content in the litter influences the ratio of nitrogen to lignin, which is a litter quality measure, and the rate of its decomposition by the microorganisms [56,57]. Apart from lignin and waxes, other polyphenolic compounds present in conifer needles are more difficult to decompose than angiosperm leaves. Phenolic compounds inhibit the enzyme activity and the precipitation of nutrient proteins [58], while the leaves of angiosperms contain water-soluble compounds such as sugars, aliphatic acids, and amino acids that are easily decomposed [59,60]. Several preliminary studies in the different ecosystems, including those developing in the urban–industry habitats, present incoherent results [61–64]. The need for research providing a better understanding of the factors determining the composition and activity of the soil microorganisms is increasing [65,66] (Figure 2).

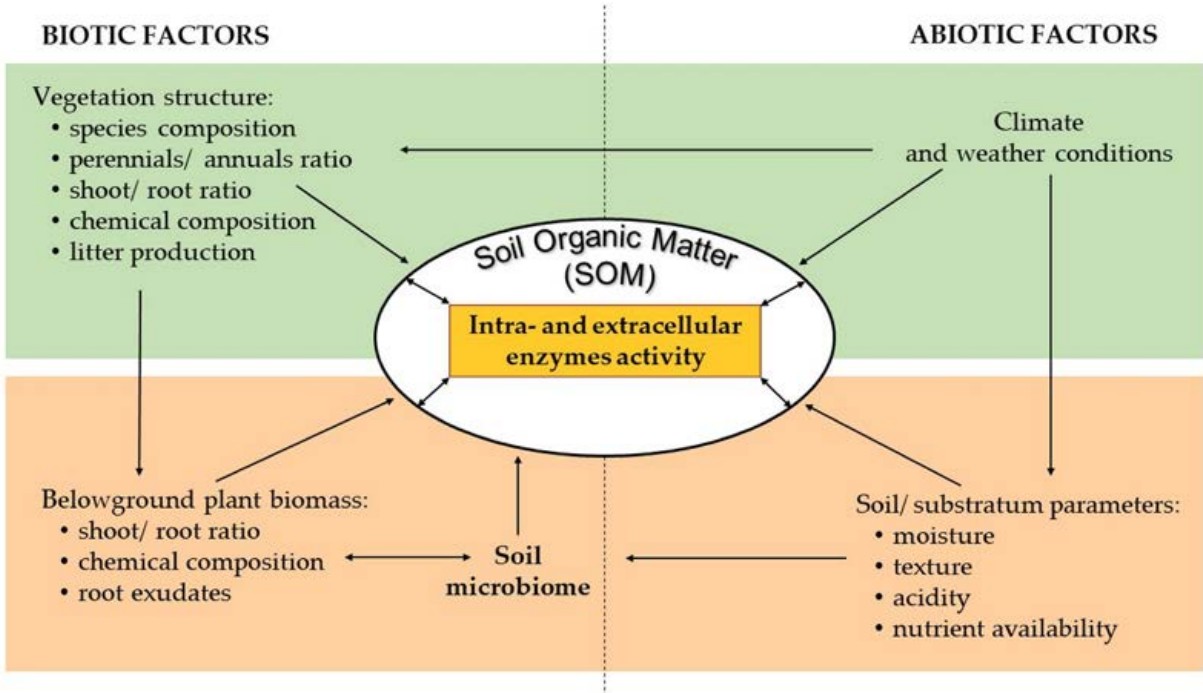

**Figure 2.** The chosen factors determining the microbiome enzymatic activity (green box–aboveground level, orange box–belowground level).

Root exudates increase the number and activity of the microorganisms in the rhizosphere [67], and their characteristics are specific for a given plant species. The composition of root exudates differs significantly among the plant species [68]. For example, American beech released more amino acids and organic acids than sugar maple, while yellow birch released more sugars than either of those two species. Moreover, the roots of the silver birch secreted a higher quantity of more diverse organic acids than the Norway spruce [69]. Among the root exudates, the low molecular carbon compounds, e.g., sugars, amino acids, and organic acids, are easily used by the microorganisms and regulate the dynamics of the soil microbial assemblages [70,71]. Root exudates stimulate bacterial growth, resulting in a more effective distribution of the environmental pollutants [72].

An important functional plant species trait influencing the soil enzyme activity is the root/shoot ratio. It exceeds 1.0 for annual plants, while perennials usually have a value lower than 1.0. Moreover, about fifty-percent of the total biomass of roots is found at depths of 0–20 cm in the soil. The assessment of the root biomass input is complex because of the continuous contact between the plant roots and the soil decomposer communities. The rhizodeposits are composed of root cap cells, organic acids secreted by plant roots, lysates of root tissues, and high-molecular-weight root mucilage.

## 4. The Physicochemical Properties Filtering the Plant Diversity and the Species Composition

Before the dynamic relationships between the plant species and the microorganisms can be analyzed, the mechanisms underlying vegetation community assembly must be understood [73]. The theory of environmental habitat filtering provides an understanding of the role of the abiotic factors in shaping the distribution of the species across a landscape habitat mosaic. It explains that the basic stage of the development of the different vegetation patches is the selection by an environmental 'filter' that permits the establishment of organisms with particular traits or phenotypes appropriate for the specific site conditions in space and time. The environmental filtering concept is followed by the study of the plant community assembly theory [74–77], succession, and biogeography [78–82].

The environmental habitat filtering theory recognizes that not all of the organisms will be able to grow under all abiotic conditions. The environmental conditions are force-selecting species that are able to tolerate a given set of abiotic and biotic conditions. According to this concept, the species assembled in the plant communities present similar phenotypic traits, reflecting the tolerance of site factors. The phenotypic similarity of the phenotypic convergence reveals the essential ecological dimensions. Phenotypic convergence has been tested previously, concerning a null model sampled from the same species pool [78,80]. Apart from the site quality, biotic interactions can impact the species composition, causing differences in success, and can also lead to shifts in the species abundance. Some species become the dominant species [83], and their functional trait identities reflect the habitat mosaic [82,84]. The environmental filtering concept reveals why studies focused on the patterns of the functional and phylogenetic assemblages, including the interactions among the plant species and soil organisms, and the need to always analyze the role of the site abiotic physicochemical parameters to explain the community species composition. The biotic factors are recognized as the realized niche of a species, while the abiotic factors reflect the characteristic fundamental niche for a given species [9,85,86]. The processes connected to the environmental filtering lead to phylogenetic and functional convergence [87–91], while the interactions between the living organisms, e.g., symbiosis, feedback relationships, and competition, drive phenotypic, phylogenetic, and functional divergence.

The niche differences stabilize coexistence and determine competitive species dominance [87]. The spectrum of species traits causes intraspecific variations that are expected to arise from species differences in resource use [89]. Competitive exclusion is strongest among pairs of similar species, causing overdispersion in the traits of the coexisting species, opposite to the assembling forces that are predicted to develop via environmental filtering [78,80,92,93].

## 5. Soil Properties, Vegetation Composition, and Soil Enzyme Activity in Novel Ecosystems

Human activity brought us to the Anthropocene Epoch. Individuals have to face the constraints of biodiversity loss, water retention needs, and balancing the elevated amount of carbon dioxide on one side. On the other side, the enormous power of the natural adaption processes leads to the establishment of non-analogous plant species composition, colonizing de novo established habitats, e.g., mineral and oligotrophic habitats, and developing them into novel ecosystems [20,94,95]. There is a need for the identification and the better understanding of the spontaneous natural processes occurring in novel ecosystems. Unfortunately, works that fulfill the prerequisites of these novel ecosystems can be counted on one hand (Figure 3).

### 5.1. Relations between Soil Physicochemical Properties and Soil Enzyme Activity in Novel Ecosystems

Some studies have revealed that enzyme activity depends on soil properties, such as moisture, pH, soil organic matter (SOM), and heavy metal content. However, there are still some doubts [96]. Replicated research has shown that enzyme activity is strongly related to the soil organic matter (SOM) characteristics. SOM provides microorganisms and their enzyme activity with energy [8]. Along with the biochemical changes recorded in the soil, the soil fertility index supports an understanding of the relationships between urease, dehydrogenase, acid, as well as alkaline phosphatase activities and soil organic carbon content [97]. The lack of correlation between the enzyme activity of a given habitat and the soil organic carbon (SOC) in spontaneous ecosystems in coal mine heaps might be caused by the presence of geogenic coal [98–103].

In the urban–industrial landscape, the traffic on roads and railways is very important. Research focusing on the enzymatic activity in soils exposed to the impacts of road traffic and heavy metal content did not prove significant relationships between SOC in terms of substrate and dehydrogenase activity or both acid and alkaline phosphatases [104].

These studies have revealed negative correlations between the urease activity and the SOC [105]. Opposite results have been obtained in analyses conducted in mixed-oak forests and vegetation chronosequences [106,107]. These issues have been an ongoing research subject in recent years (Figure 4).

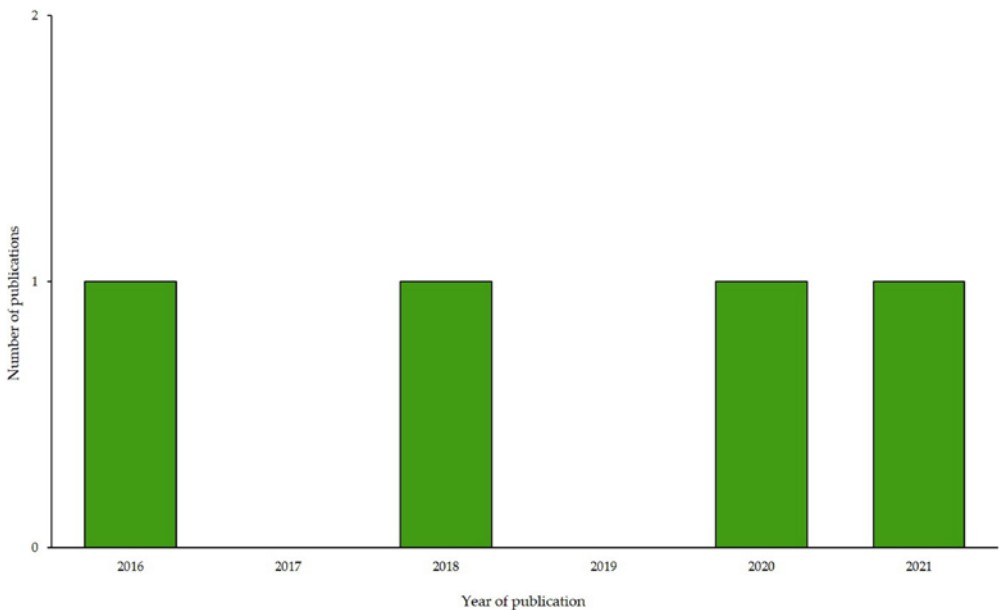

**Figure 3.** Number of papers which studied the aspects of soil microbial activity in novel ecosystems during 2016–2021.

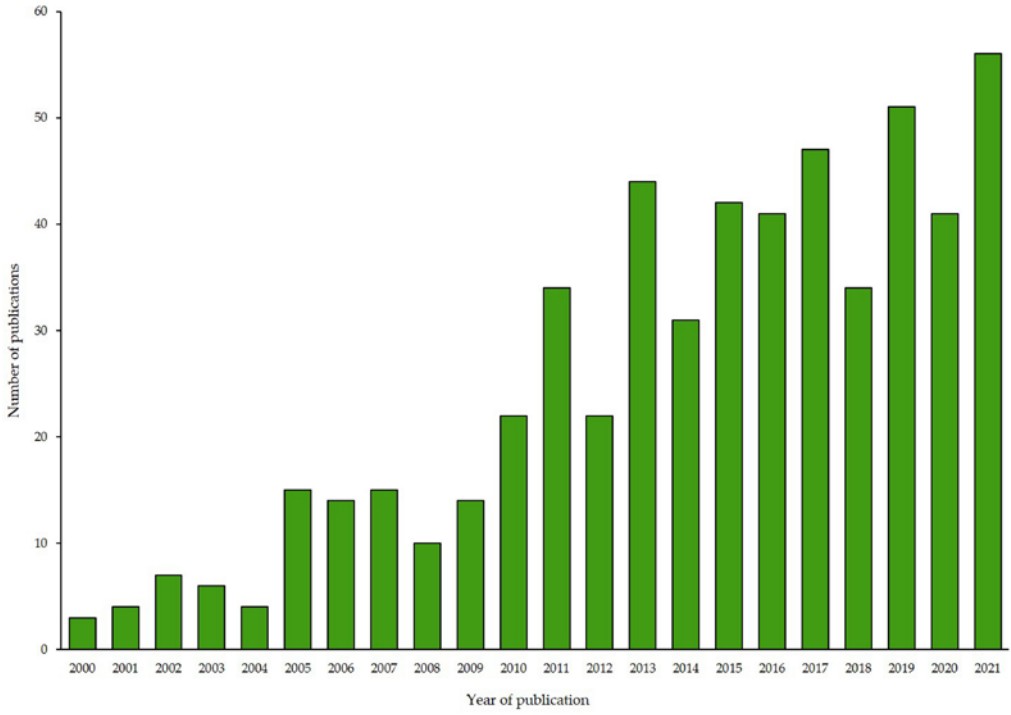

**Figure 4.** Number of papers which studied soil microbial activity in urban areas during 2000–2021.

Some studies have tested the relationship between the soil enzyme activity and the pH [107,108]. Soil pH affects the activity of soil enzymes through different mechanisms. Changes in the ionic form influence the action of enzymes, involving the affinity of substrates to the enzyme, and accounting for the decrease in enzyme activity recorded when

the pH deviates from the optimum [109]. Some studies reported that the soil enzymatic activity and the soil bacteria functional diversity increased along with increasing soil pH [107,110]. Other studies did not find correlations between the enzymatic activity, the microbial abundance, the pH, and reclamation actions (control, plantation, and mixed forests) on coal mine sites [31]. Studies conducted on the mineral substrates from coal mine heaps showed significant positive correlations between dehydrogenase and alkaline phosphatase activity and soil functional diversity and substrate pH. Both dehydrogenase and alkaline phosphatase are optimal in neutral or alkaline soil pH (7.1–10), [96,109–111].

The availability of nutrients, particularly in mineral habitats, is also critical. Research on coal mine heaps revealed positive correlations between the activity of dehydrogenases, alkaline phosphatases, and available phosphorus [61]. Similar results have been obtained from a study on lignite open-cast reclaimed heaps. The relationship between available phosphorus content and the activity of dehydrogenase was positive [112]. The high activity of soil phosphatases might be related to a low phosphorus supply for microbes [113]. However, Olander and Vitousek [114] have reported that the high content of total phosphate in the soil/substrate could be related to a decrease in the acid phosphatase activity.

Moisture and water availability in the studied habitats significantly impact the composition of the microbial assemblages, and, consequently, the soil (substrate) enzyme activity. Sufficient moisture provides soluble forms of organic matter compounds into the soil solution [14,96]. Water and moisture conditions of mineral coal mine heap habitats are complex. The porous surface of the mineral substrate causes quick drying out, which is especially significant during the summer months. The release of large amounts of sodium and the resulting encrustation of the heavily weathered slopes is commonly observed. However, the presence of montmorillonite, a mineral that can absorb a relatively high amount of water in its plate-shaped particles, can prevent the drying out [115]. In the research on the mineral habitats of coal mine heaps, positive correlations were found between the water holding capacity (WHC) as well as the substrate moisture and the soil (substrate) enzyme activity. The humidity in the samples taken from the soil substratum of coal mine heaps was relatively low in all plot types [61].

*5.2. Relations between the Vegetation and the Soil Enzyme Activity in Novel Ecosystems*

Dispersal and environmental filters have an influence on vegetation composition. In the urban–industrial landscape, the spontaneous vegetation succession process frequently occurs in unfavorable conditions concerning the soil/substrate (including low moisture and insufficient nutrient availability). Such habitat conditions have a crucial effect on the soil enzyme activity and the plant diversity [116]. Plant cover type can indirectly or directly modify the soil characteristics and properties, which can be expressed by the soil fertility index. The results obtained by Kompała-Bąba et al. [61], on hard coal heaps, proved that significantly higher enzyme activity was recorded in patches covered with vegetation than in plots without vegetation cover. Some studies indicate that the assemblages of soil microorganisms can change the microbiological processes, regardless of the habitat and the environmental factors, such as the soil temperature, water availability, and the pH [62,63]. The composition and activities of the soil microorganisms depend on factors related to the species assembled in a vegetation patch, such as the soil pH [62,63], phenolic compounds [58], or the availability of carbon and other related nutrients [63,64]. Studies on coal mine heaps and brown coal open-cast heaps revealed some regularities [29,31,61,117–121]. Vegetation composition can enhance the soil/substratum fertility directly, through its impacts on the soil/substrate organic matter (SOM) quality and quantity. Indirectly, the chemical composition of the biomass and the root exudates determines the relationship between the plants and the microorganisms and their activity [122]. A study conducted on coal mine heaps partially confirmed the above statements. Kompała-Bąba et al. [61] revealed that a higher soil fertility index is related to enzyme activity in the vegetation patches with a higher biomass input. Despite the fact that enzyme activity is

enhanced on patches with a higher plant diversity (as according to the Shannon–Wiener diversity H') and plant species richness, and higher enzyme (dehydrogenase, urease, acid phosphatase, and alkaline phosphatase) activity was observed in relation to the presence of vegetation dominated by two different functional groups: grasses and forbs (Table 1) [61].

However, some papers reported that, during the pioneer stage of succession, the relationship with the biomass input seems to be less significant in enhancing the enzyme activity in the soil/substrate [116,123]. Potentially, the abiotic stress influences the pioneer plant species, and enzyme activity is therefore less developed. During the primary stage of succession, pioneer plants, due to ecophysiological traits, can use the soil resources in various ways [116,123]. In the later stages of succession (forest communities), the properties of the soil substrate are less crucial to the microbial activity than the litter quality in the case of reclaimed open-cast lignite mines [117]. These results showed that, at this stage, the vegetation influences the enzyme activity more significantly [118]. This general trend of increased enzyme activity in the later stages of spontaneous vegetation succession was also demonstrated by other authors [29]. Additionally, it was also found that the activity of soil enzymes increases with the age of the plant community, along with the nutrient and carbon gradients [15]. According to Kara et al. [124] and Kang et al. [125], long-term afforestation can significantly increase the SOM content, accumulate the microbial biomass, and improve the potential enzyme activity [124,125]. This is still a problem that needs to be studied, regardless of the fact that the number of studies focused on the microbial activity cross-related to plant diversity have increased in the last two decades (Figure 5).

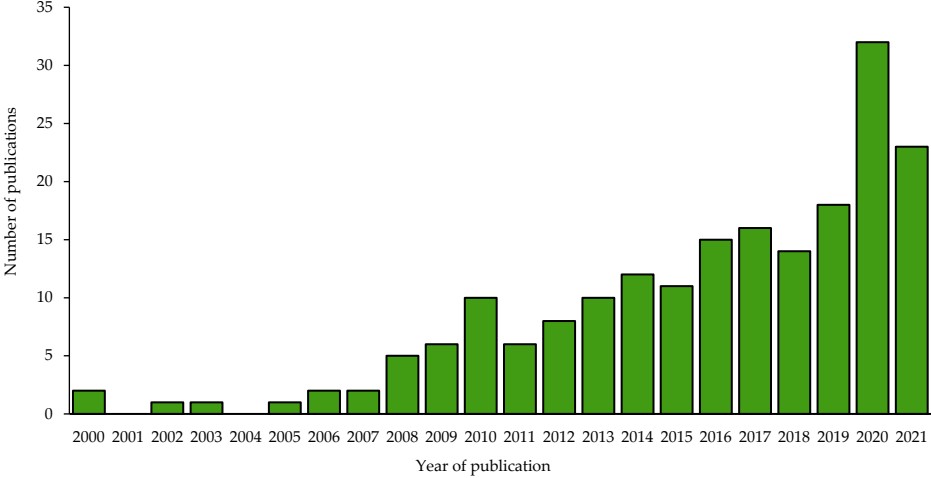

**Figure 5.** The number and publication year of papers which studied soil microbial activity in relation to plant diversity.

Study results show that the vegetation composition affects the soil biological parameters due to differences in the root exudates and the litter input among the plant species [126,127]. Research on coal mine heaps have revealed that dehydrogenase, as well as alkaline and acid phosphatases activity and the soil fertility index, are higher in the vegetation patches dominated by grasses (*C. epigejos* and *P. compressa*) compared to other vegetation types (*Daucus carota*, *Tussilago farfara*), as well as to the non-vegetated sites (Table 1).

The well-developed root system of grasses is related to the higher release of the root exudates. The chemical substances excreted by the roots affect the quality, quantity, and activity of the communities of microorganisms in the rhizosphere [6,7,128].

However, results obtained by Elhottová et al. [119] and Stefanowicz et al. [103] revealed that pioneer species such as *T. farfara* growing on hard coal mine heaps (Figure 6) caused significant increases in the diversity, the biomass, and the activity of microbial communities in these habitats. On sites with insufficient available phosphorous, such as hard coal mine

mineral habitats, a positive correlation between the diversity of plant cover and the activity of soil alkaline phosphatase has been reported [61]. The opposite result was obtained in diverse mixed-oak forests where a negative correlation between the activity of acid and alkaline phosphatases in the soil and the diversity of herbaceous plants and ferns in the substrate were found [107]. The authors explained the result by suggesting that the occurrence of competition can rule vegetation patches, so that habitats with sufficient phosphorous have a lower diversity of plant species due to competitive exclusion [107]. On the brown coal spoil heaps, a higher activity of enzymes, including cellobiohydrolase, xylosidase, and acid as well as alkaline phosphatase, was observed in soils overgrown by deciduous trees (particularly *Alnus* sp., *Tilia cordata* or *Robinia pseudoacacia*) in comparison to soils overgrown by coniferous species (particularly *Pinus* or *Picea*) [129,130], (Table 1). Deciduous trees produce litter that contains readily degradable compounds and support the development of large and active soil microbial biomass in comparison with coniferous species [131]. Therefore, a high microbial biomass under the deciduous trees explains the relatively high enzyme activities in soils under this species. On sand post-mining sites, a higher activity of soil acid and alkaline phosphatase in the deciduous stands was also detected [132], (Table 1). A high activity of phosphatases in soil overgrown by deciduous trees can also be connected with a higher N content in the soil. Previous studies have indicated that soil N content is a good predictor of phosphatase activity [133,134]. High contents of N support high phosphatase activity because the synthesis of phosphatases requires large amounts of N [134,135].

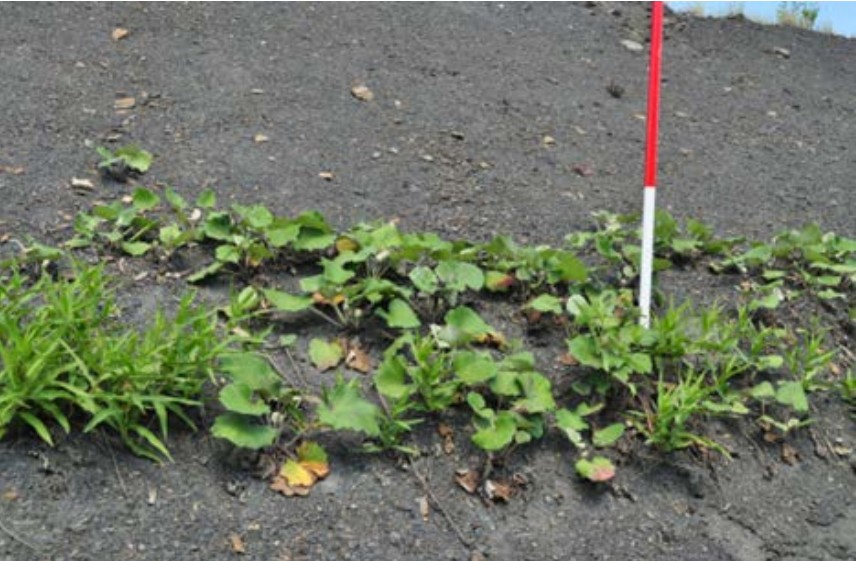

**Figure 6.** Spontaneous vegetation type with the dominance of *Tussilago farfara* (Sośnica heap, Upper Silesia, Poland)—photo: Gabriela Woźniak.

**Table 1.** Effects of different plant species on soil enzyme activity in novel ecosystems.

| Type of Site | Enzyme | Grasses | | | Herbs | | | Coniferous Trees | | | | | Deciduous Trees | | | |
|---|---|---|---|---|---|---|---|---|---|---|---|---|---|---|---|---|
| | | Calamagrostis epigejos [61,136,137] | Poa compressa [61,136,137] | Digitaria eriantha + Cynodon dactylon + Eragrostis curvula [138] | Daucus carota [61,136,137] | Tussilago farfara [61,136,137] | Lantana camara [139] | Picea omorica + Picea pungens [129] | Pinus contorta + Pinus nigra [129] | Pinus sylvestris [130,132] | Larix decidua [129] | Quercus robur [129] | Betula pendula [130,132] | Alnus glutinosa + Alnus incana [129,130,132] | Tilia cordata [129] | Robinia pseudoacacia [130] |
| Coal mine heap | Dehydrogenase | 60–120 [1] | 140–199 [1] | | 57–65 [1] | 24–26 [1] | 2.5 [1] | | | | | | | | | |
| | Urease | 0.20–0.30 [2] | 0.20–0.22 [2] | 5 [2] | 0.10–0.20 [2] | 0.10–0.20 [2] | | | | | | | | | | |
| | Acid phosphatase | 1165–1754 [3] | 899–1790 [3] | | 329–529 [3] | 221–263 [3] | | | | | | | | | | |
| | Alkaline phosphatase | 1445–3008 [3] | 1966–2751 [3] | | 718–1057 [3] | 546–1247 [3] | | | | | | | | | | |
| Brown coal heap | β-glucosidase | | | | | | | 1395 [4] | 1500 [4] | | 2297 [4] | 1500 [4] | | 2423 [4] | 1668 [4] | |
| | Celobiohydrolase | | | | | | | 582 [4] | 519 [4] | | 718 [4] | 708 [4] | | 1253 [4] | 949 [4] | |
| | Xylosidase | | | | | | | 267 [4] | 215 [4] | | 362 [4] | 225 [4] | | 834 [4] | 424 [4] | |
| | Acid phosphatase | | | | | | | | | 213 [3] | | | 234 [3] | 291 [3] | | 322 [3] |
| | Alkaline phosphatase | | | | | | | | | 218 [3] | | | 275 [3] | 284 [3] | | 248 [3] |
| Sulphur mining heap | Dehyrogenase | | | | | | | | | 100 [1] | | 690 [1] | | | | |
| Sand post mining site | Urease | | | | | | | | | 56 [2] | | | 148 [2] | 89 [2] | | |
| | Acid phosphatase | | | | | | | | | 123 [3] | | | 331 [3] | 404 [3] | | |
| | Alkaline phosphatase | | | | | | | | | 14 [3] | | | 275 [3] | 222 [3] | | |

[1] $\mu$g TFP g$^{-1}$ soil dry mass h$^{-1}$; [2] $\mu$g N g$^{-1}$ soil dry mass h$^{-1}$; [3] $\mu$g NP g$^{-1}$ soil dry mass h$^{-1}$; [4] U g$^{-1}$ soil dry mass; TPF, triphenylformazan; NP, nitrophenol; U, units.

### 6. Conclusions

The diversity of the vegetation patches, including the functional group assembly and the essential species richness parameters, as well as its impacts on the soil microbial activity and biomass, and other soil functions, have been frequently studied [49,140,141]. Soil enzyme activity is used as an indicator that can dynamically reflect the changes in the microbial community structure [142]. The occurrence of plants with particular functional traits is an essential factor influencing the activity of soil enzymes. The chemical characteristics of the plant root residues, as well as the below- and above-ground biomass, attract and shape the composition and activities of the soil microbial communities [143]. In this way, soil enzymes reflect the sum and interactions of the ecosystem structure elements very well, irrespective of the history of the vegetation. What is more important is that the soil microbial activity reveals the ability to respond quickly to changes in the environmental habitat conditions [111,121] in addition to the diversity of vegetation species composition [52,144].

The abiotic parameters of the substrates on coal mine heaps, especially the water holding capacity (WHC) and the pH, have a more significant effect on the substrate enzyme activity than the species diversity and the biomass associated with the plants at the early successional stage [61]. Further research is needed to bring the additional knowledge that could support the understanding of the factors affecting the enzyme activity concerning the variety of soil/substrate parameters and the different plant species compositions within the vegetation type, especially in novel ecosystems. Increasing our knowledge will support potential actions with which to enhance the optimal functioning and productivity of them, particularly in the oligotrophic, post-mineral excavation habitats. This knowledge will also help to create efficient tools with which to improve the habitat of the complex coal mining sites. The Anthropocene Epoch challenges the possibilities of enhancing the ecosystem re-establishment, based on the natural processes that ultimately improve the biodiversity of novel ecosystems.

The fact that the novel ecosystem has started to become established indicates that we have to change our approach. The crossing of the ecological threshold is the trigger of the intense adaptation processes in the organisms, as well as in modification of the relations between the organisms (e.g., plants with microorganisms) and the organisms with a challenging habitat. Using traditional approaches to unknown processes, humans are risking the loss of using and understanding the natural-based solution (NBS). None of the human technologies and innovations have a chance to be better than the natural adaptation solutions. Apart from agriculture and forestry, it is important to provide space for the unmanaged wildlife habitats [145]. The ecosystems developing in unmanaged habitats, including the valuable, but poor oligotrophic post-mineral excavation sites, are crucial for providing the ecosystem services for the densely populated urban–industry landscapes.

**Author Contributions:** Conceptualization, W.B. and G.W.; methodology, W.B. and B.J.; software, J.B.; validation, J.C., J.B. and A.H.; formal analysis, W.B., A.B. and A.M.J.; investigation, A.B., B.J.; resources, J.C. and G.W.; data curation, W.B., A.K.-B. and A.H.; writing—original draft preparation, W.B., J.C., A.H. and G.W.; writing—review and editing, A.K.-B., A.M.J., A.H. and G.W.; visualization, A.H. and J.B.; supervision, W.B. and G.W.; project administration, G.W.; funding acquisition, J.C., D.P. and G.W. All authors have read and agreed to the published version of the manuscript.

**Funding:** This research was funded by The National Centre for Research and Development, Grant Number: TANGO1/268600/NCBR/2015 (INFOREVITA–Geoinformatics tools, a supporting system of coal mine spoil heap reclamation); National Science Centre Poland, Grant Number: OPUS 2017/25/B/NZ8/02449 (Ocena zależności pomiędzy funkcjonalną różnorodnością roślin, strukturą zespołów mikroorganizmów i bilans węgla w czasie spontanicznej sukcesji na terenach poprzemysłowych z wykorzystaniem analiz metatranskryptomicznych); OPUS 2019/35/B/ST10/04141 (Linking soil substrate biogeochemical properties and spontaneous succession on post-mining areas: novel ecosystems in a human-transformed land-scape). Publication co-financed from the state budget under the program of the Minister of Education and Science under the name "Excellent Science"

(project number DNK/SN/551023/2022, co-financing amount PLN 213655.50, total project value PLN 238246.87).

**Institutional Review Board Statement:** Not applicable.

**Informed Consent Statement:** Not applicable.

**Data Availability Statement:** Not applicable.

**Acknowledgments:** We would like to thank Monika Malicka for providing suggestions to the manuscript, and Jacek Kasztowski for invaluable laboratory assistance during the previous study. We kindly thank Lee E. Frelich (Department of Forest Resources, University of Minnesota, USA) for the valuable comments and linguistic revision of the manuscript. We would like to thank the Mineral and Energy Economy Research Institute of the Polish Academy of Sciences for organizing the IMF and SEP which led to discussing and sharing the ideas in this manuscript.

**Conflicts of Interest:** The authors declare no conflict of interest.

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
