# Peer review of "Plant Diversity and Species Composition in Relation to Soil Enzymatic Activity in the Novel Ecosystems of Urban–Industrial Landscapes"

_sustainability, doi:10.3390/su15097284_

Round 1

Reviewer 1 Report (New Reviewer)

Dear Authors,

The article is well organized and required for present changing climate conditions, where the study will help in the mitigation of soil parameters to enhance microbial or plant diversity.  Article reviewed had general knowledge of soil problems in urban Industrial Area. Diversity of plants and soil enzymatic activity with respect to soil type.

Include the soil amendments/corrective measures to increase the plant and microbial diversity which will have more weightage for the article.

Soil health improvement is a prerequisite for sustaining soil health and microbial/crop diversity. Soil degradation is the major obstacle to the sustainability of crop production and human survival. With deteriorating climate change effects, erratic rainfall patterns, sudden increases in rainfall intensity and temperature fluctuation around the world is a serious challenge for farmers, environmentalists, and the common man. Maintenance of soil health with amendments helps to restore, revitalize and regain the soil quality. The direct and indirect effects of soil amendments on soil chemical, physical and biological properties significantly influence soil-plant-continuum, beneficial for soil health improvement, carbon sequestration, and crop yield improvement. A healthy soil in an arable system constitutes a good balance of organic and inorganic components. Such soil is usually characterized by higher biodiversity and lower concentration of inorganic and organic nutrients. Rejuvenating soils by amendments with easily available products and being environmentally safe is essential for improving soil health conditions.

Reference Article: Mickan BS, Alsharmani AR, Solaiman ZM, Leopold M and Abbott LK (2021) Plant-Dependent Soil Bacterial Responses Following Amendment With a Multispecies Microbial Biostimulant Compared to Rock Mineral and Chemical Fertilizers. Front. Plant Sci. 11:550169. doi: 10.3389/fpls.2020.550169

Author Response

Reviewer 2 Report (New Reviewer)

1.       The paper is original and of good quality.

2.       The abstract is very clearly described and comprehensive.

3.       Improve aim of the study

4.       The paper is clear and well written, and uses appropriate language.

5.       The figures and tables aren’t adequate and explain the data. The number of tables and figures used for the review is low.

6.       The references are not  recent. More recent references should be added. And check the format of each reference.

Author Response

Reviewer 3 Report (New Reviewer)

This manuscript reviews the relationships between the plant diversity, physicochemical substrate parameters and the soil enzymatic activity in novel ecosystems of the urban-industrial landscape. This study demonstrates soil enzyme activity reflects the sum and interactions of the elements of ecosystem structure. Further I think the research topic will be interesting for the readers of the Journal. The paper is well written and easy to understand.

Currently, the manuscript is just a data presentation. Some reasonable statistical analysis is applied, such as correlation analysis or effect analysis to prove the main conclusion, so that the conclusion of the manuscript is more reliable. Besides, Soil development and drought regimes also affect plant productivity and diversity. For example these articles “Soil development mediates precipitation control on plant productivity and diversity in alpine grasslands”; “Soil organic matter enhances aboveground biomass in alpine grassland under drought”.

Author Response

This manuscript is a resubmission of an earlier submission. The following is a list of the peer review reports and author responses from that submission.

Round 1

Reviewer 1 Report

The Manuscript needs to be reviewed due to grammar issues (e.g. sentence starting on line 369).

The Manuscript has numerous issues suggesting it is not ready to be considered for publishing. Some examples are listed below:

·         Appendix starts as Appendix B, it is unclear why and what happened to Appendix A

·         It seems the Authors forgot to delete some instructions from the Appendix as it contains the following text “All appendix sections must be cited in the main text. In the appendices, Figures, 422 Tables, etc. should be labeled starting with “A”—e.g., Figure A1, Figure A2, etc....”

·         It isn’t clear why are some Figures given more than once (e.g. Fig. 1 on the beginning and end of the paper)

·         Why does the Appendix reefer to the Figure as A1 and A2 when there are no such Figures in the Article

·         I think the referencing is off as the Authors used [x-y] notation both for subsequent numbers and intervals

In addition to the listed issue I do not understand the goal and contribution of the submitted paper. What research did the Authors implement, what are their findings, what is the novelty presented in the Manuscript?

Reviewer 2 Report

The manuscript has reviewed the important and interesting topic. In my opinion thereview is comprehen sive and has covered all important Topic. However, as a reader I would like to made some minor suggestions for consideration. 

1. Because of this is a review paper, it is importance for the reader to easily identify which statement is the opinion/summary suggested by the authors or the information form the reference(s). Therefore, I suggest the authors to carefully add the references to all sentences which are not the oppinion/summary of the authors, e.g., 

- line 120: ... reported from semi-natural alphine meadows.

line 151: ... Or water availability

2. The criteria and the method for selecting the papers is one of the important part of the review paper. I suggest the author provide this data inanother topic instead of placingin the caption of the figure.

Reviewer 3 Report

In this manuscript, the authors reviewed the relationships between the diversity of vegetation and the soil enzymatic activity in novel ecosystems of the urban-industrial landscapes. The soil enzyme activity can reflect the primary producers and the processes of matter and energy flow, which are the basics of all ecosystem's functioning processes. Moreover, the soil enzyme activity is an indicator reflecting changes in the microbial community structure. However, the factors driving the soil enzyme activity are not fully understood. Therefore, it is very important to study the influence of the soil enzyme activity in the novel ecosystems. This manuscript is well written and organized. This study can be interesting to a very broad range of researchers. So I would recommend that the manuscript be accepted.

Reviewer 4 Report

The review provides ideas on the currently available information related to the importance of enzymatic activity to the vegetation pattern in the natural and urban landscape. The edited version is clear and informative.

Round 2

Reviewer 1 Report

The submitted paper does not contain elements required to be considered a research paper.

Author Response

Dear Reviewer,
Giving a point-by-point response to the reviewer's comments is challenging as there are no points. Such criticism has to be explained.
There are no comments from the three other reviewers indicating they are probably happy with the Authors response. In the opinion of the three other reviewers, the paper contains elements required to be considered a research paper.

Dear Editor,
In the first round, we got the information that the paper is accepted after revision. How is the paper worse after revision and is rejected? We suspect that the first reviewer is not objective. There are no listed points to be improved. How is it possible to give such a revision it is not trustful? Please give us the name of the reviewer. He declared that he would like to sign the review report. We put much work into improving the paper, and we booked the money to pay for the paper. The resubmission means that we will lose the money at the end of the year.
With Professor Damian Chmura, we have worked for two years on a special issue, "The Provision of Ecosystem Services in Response to Habitat Change" for Sustainability by the Editor (winnie.cheng@mdpi.com). This paper is part of the special issue. We managed to convince the researchers working on the natural processes in the urban-industrial landscape to prepare papers for these issues to be consistent with the scientific idea. We would be very grateful if you could give us suggestions on improving the paper to get it published. We are very sorry, but we need an urgent response. We are in a hurry because of the booked money.